# Integrating In Silico and In Vitro Approaches to Identify Natural Peptides with Selective Cytotoxicity against Cancer Cells

**DOI:** 10.3390/ijms25136848

**Published:** 2024-06-21

**Authors:** Hui-Ju Kao, Tzu-Han Weng, Chia-Hung Chen, Yu-Chi Chen, Yu-Hsiang Chi, Kai-Yao Huang, Shun-Long Weng

**Affiliations:** 1Department of Medical Research, Hsinchu MacKay Memorial Hospital, Hsinchu City 300, Taiwan; 2Department of Medical Research, Hsinchu Municipal MacKay Children’s Hospital, Hsinchu City 300, Taiwan; 3Department of Dermatology, MacKay Memorial Hospital, Taipei City 104, Taiwan; 4National Center for High-Performance Computing, Hsinchu City 300, Taiwan; 5Department of Medicine, MacKay Medical College, New Taipei City 252, Taiwan; 6Institute of Biomedical Sciences, MacKay Medical College, New Taipei City 252, Taiwan; 7Department of Obstetrics and Gynecology, Hsinchu MacKay Memorial Hospital, Hsinchu City 300, Taiwan; 8Department of Obstetrics and Gynecology, Hsinchu Municipal MacKay Children’s Hospital, Hsinchu City 300, Taiwan

**Keywords:** anticancer peptide, antitumor peptide, anticancer activity, selective cytotoxicity, in silico analysis, in vitro experiments, machine learning

## Abstract

Anticancer peptides (ACPs) are bioactive compounds known for their selective cytotoxicity against tumor cells via various mechanisms. Recent studies have demonstrated that in silico machine learning methods are effective in predicting peptides with anticancer activity. In this study, we collected and analyzed over a thousand experimentally verified ACPs, specifically targeting peptides derived from natural sources. We developed a precise prediction model based on their sequence and structural features, and the model’s evaluation results suggest its strong predictive ability for anticancer activity. To enhance reliability, we integrated the results of this model with those from other available methods. In total, we identified 176 potential ACPs, some of which were synthesized and further evaluated using the MTT colorimetric assay. All of these putative ACPs exhibited significant anticancer effects and selective cytotoxicity against specific tumor cells. In summary, we present a strategy for identifying and characterizing natural peptides with selective cytotoxicity against cancer cells, which could serve as novel therapeutic agents. Our prediction model can effectively screen new molecules for potential anticancer activity, and the results from in vitro experiments provide compelling evidence of the candidates’ anticancer effects and selective cytotoxicity.

## 1. Introduction

Cancer remains a significant global health burden in the 21st century, ranking among the leading causes of death alongside cardiovascular diseases in many countries [1]. Annually, tens of millions of individuals receive cancer diagnoses worldwide, and nearly half of them succumb to the disease [2]. While surgical resection is a traditional and effective therapy for many cancer types, various additional treatments have been developed to reduce cancer cell growth and progression. These include radiation therapy, chemotherapy, immunotherapy, hormone therapy, targeted therapy, and other approaches [3]. Among them, chemotherapy, commonly known as “chemo”, is the most prevalent treatment that uses drugs to slow cancer growth or eliminate cancer cells [4,5]. Multiple chemotherapy drugs are in clinical use today due to the tendency of cancer cells to grow and divide faster than healthy cells, making them ideal targets for these drugs. However, chemotherapy often damages non-cancerous cells, causing side effects like fatigue, hair loss, lung tissue damage, cardiac and renal problems, peripheral neuropathy, and infertility [6].

Anticancer peptides (ACPs), typically consisting of 10–50 amino acids, are bioactive molecules that induce cytotoxicity against cancer cells by disrupting and penetrating cell or organelle membranes. Their selective cytotoxicity has been observed in various cancers, positioning them as potential novel antineoplastic agents [7]. One notable difference between tumor and healthy cells lies in the membrane’s electrical properties. Tumor cells secrete large amounts of lactate through glucose and glutamine metabolism, which results in a negatively charged surface [8,9]. ACPs leverage this difference to disrupt cancer cell membranes via electrostatic interactions with their anionic components, thereby selectively lysing cancer cells [10,11,12]. Compared to antibodies and small molecules, ACPs are increasingly viewed as effective and safer alternatives to chemotherapy, offering high selectivity, penetration, and easy modification. With cancer remaining a leading cause of death globally, ACPs are attracting attention for their clinical potential as a new class of antineoplastic drugs.

In recent years, in silico approaches have been employed to identify peptides with cytotoxicity against cancer cells. In 2013, Tyagi et al. introduced the AntiCP [13] model, which discriminates between ACPs and non-ACPs based on limited available data. Hajisharifi et al. later improved ACP prediction by combining Chou’s pseudo-amino acid composition with other sequence features [14]. Vijayakumar and Lakshmi developed a novel feature encoding method that identifies apoptotic domains in a peptide, enhancing sensitivity in ACP detection [15]. Chen et al. created the iACP web tool, using a feature selection algorithm to identify key features for ACP prediction [16]. Li and Wang investigated the correlation between anticancer activity and amino acid sequence properties, including amino acid composition, average chemical shift, and reduced amino acid composition [17].

Other researchers have developed models using genetic algorithms (GAs), SMOTE (Synthetic Minority Oversampling Technique), and support vector machines (SVMs) to enhance ACP prediction [18,19]. Wei et al. created ACPred-FL [20] using the minimum redundancy maximum relevance (mRMR) method to select informative features, significantly improving predictive performance. ACPred [21] used SVMs coupled with amino acid and amphiphilic pseudo-amino acid compositions, revealing that hydrophobic residues in the α-helix and cysteine residues in the β-sheet structures correlate with anticancer activity. mACPpred [22] was created based on selected physicochemical and compositional properties. Recently, AntiCP 2.0 [23], a refined version of the original model, was trained using amino acid composition and an ETree classifier, achieving state-of-the-art accuracy.

Despite numerous methods for ACP identification, comprehensive evaluation via in vitro or in vivo cytotoxicity assays remains limited. Therefore, a robust analysis platform is needed that provides high-accuracy prediction and experimental validation. In this study, we aim to develop a strategy for identifying natural peptides with cytotoxicity against cancer cells through a combined computational and experimental approach.

## 2. Results

The workflow, outlined in Figure 1, includes several key steps: data collection and preprocessing, analysis of ACP features, construction of a novel prediction model, evaluation of the model’s performance, identification of natural candidate ACPs through multiple predictive tools, and validation of ACPs’ selective cytotoxicity against specific cancer cells. The details of each step are described below.

### 2.1. Data Collection and Preprocessing of Peptides with Anticancer Activity

As summarized in Table 1, a total of 1462 experimentally verified ACP sequences were gathered from the literature [14,20,22,24] and public tools and databases including ACPred [21], ACPred-FL [20], AntiCP [13], AntiCP2 [23], APD3 [25], CAMP [26], CancerPPD [27], DADP [28], dbAMP [29], DRAMP [30], EnACP [24], mACPpred [22], and SATPdb [31]. An additional 2875 non-ACP sequences were sourced from existing tools for predicting ACPs, such as ACPred [21], ACPred-FL [20], AntiCP [13], AntiCP2 [23], EnACP [24], and mACPpred [22].

To prevent overfitting, where the model becomes too well-suited to the training data, redundant peptide sequences were removed from both positive and negative datasets. Only sequences between 10 and 50 residues long were retained. The remaining dataset comprised 804 ACP sequences and 1494 non-ACP sequences, resulting in a 1:2 ratio of positive to negative sequences. At the time of analysis, this was the most comprehensive data available.

To identify potential ACPs derived from natural sources, peptides ranging from 10 to 50 amino acids in length without an anticancer activity annotation were extracted from the Universal Protein Knowledgebase (UniProtKB) [32], resulting in 41,489 sequences used as the testing dataset.

### 2.2. Investigation of Sequence and Structural Features of Anticancer Peptides

Several studies have shown that sequence-based features are highly effective for predicting protein functions. In this research, amino acid composition (AAC) [33], dipeptides composition (DPC) [34], and k-spaced amino acid pairs (CKSAAPs) [35] were employed to differentiate ACPs from non-ACPs. After preprocessing, the occurrence frequencies of the 20 amino acids were calculated to identify consensus motifs in ACP sequences. Figure 2 compares the composition of essential amino acids between ACP and non-ACP sequences, revealing that aliphatic residues glycine (G) and leucine (L) are enriched in ACPs. Aromatic amino acids phenylalanine (F) and tryptophan (W) are also more frequent in ACPs compared to non-ACPs. Notably, lysine (K), a basic amino acid, shows the most statistically significant difference in frequency. This indicates that ACPs primarily interact with cancer cells through electrostatic interactions with anionic phospholipids in the plasma membrane. This is a key mechanism through which ACPs disrupt membrane integrity, leading to the leakage of cellular contents [7,36].

Additionally, cysteine (C), an amino acid with both polar and hydrophobic properties, plays a critical role in protein structure and stability and is more frequently found in ACPs. Studies have shown that cysteine-rich, cationic peptides with antimicrobial activity also exhibit cytostatic effects against cancer cells [37,38,39,40]. Many of these ACPs, including defensins and bacteriocins, have been reported to show low cytotoxic and hemolytic effects on normal cells [41]. Conversely, amide and acidic amino acids like asparagine (N), glutamine (Q), aspartic acid (D), and glutamic acid (E) are polar and negatively charged at physiological pH. The results show that these amino acids are less prevalent in ACPs.

Analyzing amino acid pairs helps estimate the significance of different combinations and their characteristics. For each peptide sequence, the composition of amino acid pairs was measured at k-spaced intervals of zero, one, two, and three residues. Figure 3 shows the frequency differences of 400 k-spaced amino acid pairs between ACPs and non-ACPs using 20 × 20 matrices, highlighting enriched and suppressed pairs in red and green, respectively. At zero spacing (k = 0), the pairs correspond to dipeptides and are enriched with aliphatic (G, A, V, I, L) and basic (K, R, H) amino acids, such as AK, RR, GG, GL, GK, LA, LL, LK, KA, KI, KL, and KK. When k = 1, pairs like AxK, GxL, IxK, LxK, KxA, KxL, and FxK are significantly different between ACPs and non-ACPs. At k = 2, pairs such as AxxK, GxxC, LxxL, LxxK, KxxA, KxxL, and KxxK show marked differences. At k = 3, the pairs AxxxA, CxxxC, GxxxK, LxxxA, LxxxL, KxxxK, and FxxxL are enriched in ACPs. The presence of sulfur-containing cysteine in various combinations across all k-spacings indicates the importance of these pairs for distinguishing ACPs.

Protein folding, which involves the number, spatial arrangement, and connectivity of secondary structure elements, plays a crucial role in biological functions [42]. Using the PEP2D tool [43], the secondary structure elements composition (SSEC) was predicted for each peptide. Figure 4 compares the secondary structure compositions between ACPs and non-ACPs, revealing that ACPs are composed of 56.9% random coils, 31.9% alpha-helices, and 11.2% beta-strands. The results show a significant difference in coil structures between ACPs and non-ACPs, with ACPs showing fewer helices and more beta-strands. Furthermore, the SSEC of the first and last 10 residues were analyzed separately. The C-terminus of ACPs contains more beta-strands and fewer helices and coils compared to non-ACPs, while no significant difference was found at the N-terminus. Studies investigating ACP structure and biological activity [44,45,46,47,48] suggest that the alpha-helical structure plays a crucial role in the anticancer effects and selective cytotoxicity of ACPs against cancer cells [49,50,51].

Finally, the amino acid and structural element compositions at the N- and C-terminus of ACPs and non-ACPs were compared using the TwoSampleLogo version 1.21 [52]. Figure 5 shows position-specific AAC and SSEC for the first and last five amino acids. Positively charged amino acids lysine (K) and arginine (R) are enriched at the C-terminus of ACPs, while nonpolar residues like phenylalanine (F), leucine (L), tryptophan (W), and proline (P) are particularly abundant at the N-terminus. Previous studies [13,53] suggest that the positively charged C-terminus plays a significant role in affecting tumor growth and progression. In addition, position-specific SSEC analysis indicates a greater prevalence of C-terminal beta-strands in ACPs compared to non-ACPs.

Overall, the amino acid compositions and conformations of the C-terminal region appear to play a critical role in determining a peptide’s ability to suppress cancer cells.

### 2.3. Construction of Prediction Models Based on Sequence and Structural Features

To evaluate the discrimination capability of the investigated features for distinguishing ACPs from non-ACPs, we trained models using each feature subset and validated them through five repetitions of five-fold cross-validation. Each peptide sequence was encoded using different feature encoding methods, including AAC, DPC, C1SAAP, C2SAAP, C3SAAP, SSEC, N-AAC, N-SSEC, C-AAC, and C-SSEC. The LIBSVM tool [54] was used to build the SVM prediction models. Table 2 presents the results, where the AAC model achieved satisfactory results, with a sensitivity of 89.00%, specificity of 89.48%, accuracy of 89.31%, and a Matthews correlation coefficient (MCC) of 0.77 in distinguishing ACPs from non-ACPs. The models based on CKSAAP features demonstrated exceptional performance, with the C1SAAP model delivering the best results, showing a sensitivity of 90.00%, specificity of 90.09%, accuracy of 90.06%, and an MCC of 0.79.

Unfortunately, the model trained using SSEC features could not effectively distinguish ACPs from non-ACPs, resulting in suboptimal performance, with a sensitivity of 62.29%, specificity of 64.08%, accuracy of 63.46%, and an MCC of 0.25. Additionally, the models trained on N- or C-terminal amino acid or secondary structure element compositions also yielded subpar sensitivity values (all below 70%) except for the N-AAC model, which performed slightly better.

These findings suggest that sequence-based features are valuable for characterizing peptides with anticancer activity. However, secondary structure elements provide limited predictive power, possibly because they were approximated as substitutions.

### 2.4. Performance Evaluation of Model Trained by Hybrid Feature Sets

Based on previous results, models trained using sequence-based features demonstrated efficient performance in classification on the training dataset. However, according to prior research [22,23,55], models that incorporate hybrid feature sets generally achieve higher average accuracy than those utilizing individual features. Consequently, to enhance predictive capability, these features were combined both additively and in a more integrated manner and applied to the SVM classifier.

As depicted in Table 3, the models that integrated sequence and structural characteristics showed improved performance. The model utilizing a combination of AAC and DPC features achieved a sensitivity of 91.02%, a specificity of 90.12%, an accuracy of 90.44%, and an MCC of 0.80. Remarkably, the model combining AAC, DPC, and CKSAAP delivered the best overall performance, with a sensitivity of 91.17%, a specificity of 90.83%, an accuracy of 90.95%, and an MCC of 0.81. Although combining multiple sequence-based features enhanced classification performance, models that integrated both sequence and structural features still showed less satisfactory results. Specifically, the model combining AAC, DPC, and SSEC yielded slightly reduced predictive performance, with a sensitivity of 84.3%, specificity of 85.07%, accuracy of 84.8%, and an MCC of 0.68. Similar results were observed when adding CKSAAP and SSEC to the combination.

Five-fold cross-validation was used, and Figure 6 illustrates the comparison of receiver operating characteristic (ROC) curves between the SVM models trained using all feature sets. The area under the ROC curve (AUC) for each model was measured. In summary, the model trained by combining sequence-based features such as AAC, DPC, and CKSAAP significantly enhances the predictive performance for distinguishing between ACPs and non-ACPs.

### 2.5. Identification of Natural ACPs by Integrating Multiple Tools

Antimicrobial peptides have been discovered across a broad range of life forms; however, only a small number of peptides with anticancer activity have been identified and validated through biological experiments. We propose a strategy for identifying anticancer peptides derived from the natural environment, which may offer cancer treatment with fewer complications. To achieve more precise identification, various approaches were used to predict anticancer activity from the natural peptide dataset sourced from UniProtKB [32]. These tools include ACPred [21], ACPred-FL [20], AntiCP [13], AntiCP2 [23], iACP [16], mACPpred [22], and our proposed model.

As outlined in the methods section, a peptide was considered a candidate if it was predicted as a positive case by all the aforementioned tools. In total, 176 natural peptides out of 41,489 were considered potential ACP candidates (Appendix A). Table 4 lists the top 20 candidates with the highest probability, with many of these peptides originating from plants, notably from species such as Oldenlandia affinis (OLDAF), Chassalia parviflora (CHAPA), Psychotria brachyceras (PSYBR), and Psychotria leiocarpa (PSYLE) in the Rubiaceae family, as well as Viola odorata (VIOOD), Viola hederacea (VIOHE), Viola inconspicua (VIOIN), Melicytus dentatus (MELDN), and Melicytus chathamicus (MELCT) in the Violaceae family. Additional candidates were identified in amphibians from the Viperidae family, such as Crotalus durissus ruruima (CRODR), Crotalus viridis (CROW), and Crotalus durissus terrificus (CRODU), and in species of tree frogs like Phyllomedusa trinitatis (PHYTB) and Ranoidea caerulea (RANCA), as well as in bees and bacteria.

Furthermore, functional enrichment analysis was performed to identify the biological themes present in the candidate ACPs. As shown in Figure 7, the results indicated that these candidates were significantly enriched in Gene Ontology (GO) [56] terms related to defense responses to bacteria and fungi, the killing of cells from other organisms, cell cytolysis, degranulation, and hemolysis within the biological process (BP) category. The significantly enriched GO terms in the cellular component (CC) category included extracellular region and membrane, and for molecular function (MF), the enriched terms involved toxin and hormone activity. These findings suggest that these peptides may play a crucial role in regulating defense responses against various pathogens, likely aiding in the fight against harmful bacteria and cancer cells.

### 2.6. Validating the Selective Cytotoxicity of ACPs against Specific Cancer Cells

Although the anticancer activity of some peptides has been confirmed through in vitro experiments, previous studies often evaluated the cytotoxic effects in only a few cell lines, without exploring the underlying mechanisms of action in depth. To validate the anticancer activity and selective cytotoxicity of the predicted ACPs, we synthesized and tested ten putative ACPs, labeled 1 to 10, for their effects on inhibiting cell proliferation using over 30 cancer cell lines. This evaluation employed the MTT colorimetric assay and included a diverse array of cancers such as skin, lung, colon, liver, breast, stomach, endometrial, ovarian, hypopharyngeal, lymphoma, pancreatic, fibrosarcoma, prostate, brain, oral cavity, and bone cancer, as summarized in Table 5. Additionally, two peptides, numbered 11 and 12, which were highly ranked in our model but not in others, were synthesized for comparative analysis.

As depicted in Figure 8 and Appendix A, the results confirmed the anticancer effects and selective cytotoxicity of these putative peptides against human cancer cells. The half-maximal inhibitory concentration (IC50) was calculated to measure the inhibitory capacity. Peptides 1 to 5 demonstrated broad cytotoxicity against various cancer cell lines even at concentrations below 50 μM, impacting cells such as A431, H1299, A549, HT29, HepG2, HEC-1-A, FaDu, HL-60, Daudi, Panc-1, and DU145. Peptides 6 and 7 showed notable anticancer effects, particularly on liver and breast cancers, and also affected endometrial and hypopharyngeal cancer cells. Peptide 8 was crucial in selectively targeting SKOV-3 ovarian carcinoma, HT1080 fibrosarcoma, DU145 prostate cancer, and DBTRG brain tumor cells. Peptide 9 was particularly effective against liver cancer cells, especially HepG2 and Mahlavu, while Peptide 10 showed pronounced cytotoxicity in BT474 cells. Interestingly, the DBTRG cell line was relatively resistant to these ACPs, except for Peptide 11, which exhibited specific cytotoxicity against brain tumor cells at a concentration of 32.12 μM. Peptide 12 also demonstrated anticancer activity not only against the triple-negative breast cancer cell line MDA-MB-231 but also in the Burkitt lymphoma-derived Daudi cell line.

Additionally, we have generated dose–response curves for the combination of each ACP and the cancer cell line that exhibited the most significant effect, as shown in Appendix A. The anticancer mechanisms and effects of ACPs can differ significantly across various cancer cell lines due to the unique biological characteristics and microenvironments of these cells. Many studies have demonstrated that ACPs can exert a range of anticancer activities, such as inhibiting cell migration, suppressing angiogenesis, displaying antioxidant properties, halting cell proliferation, inducing apoptosis, and exerting cytotoxic effects [57,58]. This diversity in action mechanisms results in ACPs having selective efficacy against different types of cancer cells.

For instance, certain ACPs may show higher selectivity for cancer cells with a highly negatively charged cell membrane due to stronger electrostatic interactions between the peptides and the cancer cell membrane, leading to the targeted disruption of cancer cell membranes and the subsequent induction of cell death through necrosis or apoptosis [58]. Furthermore, the anticancer efficacy of ACPs is influenced by their amino acid composition, structural properties, hydrophobicity, and amphipathic nature, which enhance their interactions with cancer cell membranes [59].

In summary, the functional diversity and adaptability of ACPs result in varying levels of effectiveness in different cancer treatments, underscoring their potential for targeted cancer therapies. However, further research is necessary to elucidate the mechanisms driving these reactions. Importantly, we have shown that our model can more accurately predict peptides that possess anticancer activity.

## 3. Discussion

In this study, we merged mathematical modeling with in vitro experiments across various cancer cell lines to investigate the anticancer activity and selective cytotoxicity of peptides. This approach provided a strategic framework for researchers to identify and characterize anticancer peptides derived from natural sources. We assembled the largest collection of experimentally validated ACPs in our training dataset compared to previous studies. The analysis of sequence features offered insights into the putative functions of ACPs, particularly through the comparison of k-spaced amino acid pairs between ACPs and non-ACPs. The cross-validation results from the training dataset confirmed the effective discrimination capability of the investigated features, with some models achieving accuracies over 90%.

Furthermore, by integrating the predictions from our model with other bioinformatics tools, we enhanced the accuracy and reliability of identifying potential ACPs. The functional enrichment analysis suggested that most of the predicted ACPs play a crucial role in targeting and destroying harmful cells. As noted, the anticancer effects and selective cytotoxicity of these peptides were substantiated through in vitro experiments involving numerous human cancer cell lines. Although the mechanisms behind the anticancer properties of these peptides are not yet fully understood, our findings underscore their potential to exhibit cytotoxicity against cancer cells, emphasizing the need for further exploration to fully elucidate their therapeutic capabilities.

## 4. Materials and Methods

### 4.1. Redundancy Removal

To minimize overestimation, the CD-HIT V4.8.1 [60] was employed to decrease redundancy in the ACP sequences within the training dataset by applying a cutoff of 80% similarity. Additionally, to better mimic real-world conditions, homologous sequences in the non-ACP dataset were also filtered out when their identity exceeded 50%. This approach ensures a more accurate and reliable dataset for subsequent analyses.

### 4.2. Feature Investigation

Sequence-based features, including amino acid composition (AAC) [33], dipeptide composition (DPC) [34], and k-spaced amino acid pairs (CKSAAPs) [35], are widely utilized in analyzing protein functions and developing prediction models. AAC specifically quantifies the frequency of occurrence of each of the 20 standard amino acids in a protein sequence, which is essential for feature encoding. This process can be described as follows:$$f\left(i\right)=\frac{\sum x_{i}}{L}(1\leq i\leq 20)$$

Given a peptide, where *i* represents each type of amino acid, $x_{i}$ stands the number of occurrences of each amino acid, and *L* is the full length of the considered peptide.

The CKSAAP method estimates the frequencies of occurrence of 20 × 20 types of amino acid pairs in a peptide. These pairs are defined by their separation through a specific number of other amino acids, known as the gap. The formulation can be expressed as follows:$$f\left(i,j\right)=\frac{\sum x_{i,j}}{L-k}\;(1\leq i,j\leq 20)$$
where *i*,*j* represents each amino acid pair, $x_{i,j}$ stands the number of occurrences of each amino acid pair separated by *k* amino acids, and *L* is the full length of the considered peptide; *k* = 0, 1, 2, and 3 were considered as features to be applied in the prediction of anticancer activity.

Additionally, the composition of secondary structure elements (SSECs) was considered a primary feature for investigation in this study. The secondary structures of all peptides were predicted using the PEP2D web tool [43]. This tool uniquely predicts the secondary structure of peptides based on their amino acid sequences using a method tailored specifically for peptides rather than proteins.

### 4.3. Construction of Prediction Models

The support vector machine (SVM) is a supervised machine learning algorithm widely applied to various biological classification problems. In this study, the SVM algorithm implemented in LIBSVM [54] was used as the classifier. LIBSVM is a publicly available SVM tool that employs the radial basis function (RBF) as its kernel function. The flexibility of the decision boundary, or hyperplane, is determined by two parameters: gamma (γ) and cost (C). We utilized LIBSVM to construct classification models using feature vectors based on both sequence and structural characteristics.

### 4.4. Performance Evaluation

In this study, we used the training data to build the ACP prediction model employing LIBSVM. To evaluate the model’s performance, we conducted five repetitions of a five-fold cross-validation procedure. We utilized the following measures to estimate the predictive performance of the model, encompassing *TP* (true positive), *FN* (false negative), *TN* (true negative), and *FP* (false positive):$$Sensitivity\;(Sn)=\frac{TP}{TP+FN}$$
$$Specificity\;(Sp)=\frac{TN}{TN+FP}$$
$$Accuracy\;(Acc)=\frac{TP+TN}{TP+FP+TN+FN}$$
$$Matthews\;Correlation\;Coefficient\;(MCC)=\frac{\left(TP\times TN\right)-\left(FP\times FN\right)}{\sqrt{\left(TP+FP\right)\left(TP+FN\right)\left(TN+FP\right)\left(TN+FN\right)}}$$

### 4.5. Identification of Candidate ACPs Using Multiple Prediction Tools

Novel natural ACP candidates were predicted using the proposed model alongside six other existing models, including ACPred [21], ACPred-FL [20], AntiCP [13], AntiCP2 [23], iACP [16], and mACPpred [22]. Peptide sequences extracted from natural species were input into these prediction tools in FASTA format. By integrating the results from all seven prediction tools, we were able to obtain multiple decision-making outcomes. A majority voting method was adopted to synthesize these results into a final decision. A peptide was nominated as a candidate only if it received unanimous approval from all seven tools. If candidates obtained an equal number of votes, they were ranked based on the scores from the proposed model.

### 4.6. Functional Enrichment Analysis

Gene Ontology (GO) [56] is a comprehensive resource that describes the functions of gene products across all living species in three independent categories: cellular component (CC), molecular function (MF), and biological process (BP). To provide a functional interpretation for the identified peptides, GO annotations for each peptide were obtained from the Universal Protein Knowledgebase (UniProtKB) [32].

### 4.7. Evaluation of Anticancer Activity and Selective Cytotoxicity

Cancer cell lines were purchased from BCRC (Hsinchu, R.O.C) and their culture conditions were created according to BCRC’s suggestion. The culture medium used for the cancer cell lines included 10% fetal bovine serum (Gibco, Grand Island, NY, USA) and 1% penicillin/streptomycin (Gibco) in 5% CO_2_ at 37 °C. Cells were seeded into 96-well tissue culture plates at a concentration of 1 × 10^4^ cells per 200 μL per well and allowed to settle overnight. The cells were then treated with serial dilutions of various ACPs. After 48 h of incubation, cell viability for each line was assessed using the MTT colorimetric assay (Sigma-Aldrich, St. Louis, MO, USA). The peptide preparation and dilution in our anticancer assays are as follows: ACPs (10 mg) were dissolved in 100 μL of DMSO to generate 100 mg/mL stock solutions. These stock solutions were then diluted 100 times with a culture medium to obtain a 1 mg/mL solution. Finally, serial dilutions were prepared in a culture medium containing 1% DMSO to achieve the desired ACP concentrations for the MTT assay. Cell viability was expressed as a percentage of the untreated control, and the inhibitory concentration at which 50% of the cells survived (IC50) was determined from the dose–response curve.

## 5. Conclusions

The findings of this study significantly advance the development of cancer therapies by identifying potential anticancer peptides (ACPs) through both computational predictions and experimental validations. These ACPs exhibit selective cytotoxicity towards cancer cells, presenting a promising alternative to traditional treatments like chemotherapy and surgery, which often come with severe side effects. The discovery of these natural peptides with anticancer properties opens the door to novel, targeted therapies that are both effective and have fewer side effects.

Moreover, our study demonstrates the effectiveness of integrating computational and experimental approaches, which pave the way for more efficient discovery and validation processes for ACPs in future research. This dual approach not only increases the likelihood of discovering potent therapeutic agents but also positions ACPs as excellent candidates for the development of new anticancer drugs. By leveraging both computational predictions and empirical validations, we enhance the potential to identify effective treatments that could play a crucial role in future cancer therapies.

In conclusion, the strategy outlined in this study, which integrates in silico and in vitro approaches, offers a comprehensive and reliable platform for the identification and characterization of natural anticancer peptides. This integrated strategy not only facilitates the discovery of potent therapeutic agents but also establishes a robust framework for translating these promising peptides into effective cancer treatments.

## Figures and Tables

**Figure 1 ijms-25-06848-f001:**
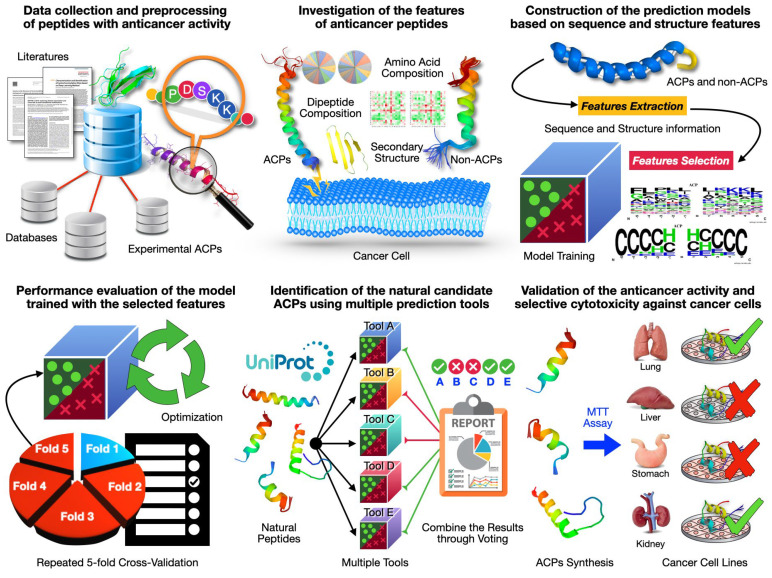
Flowchart depicting process for identifying natural peptides with selective cytotoxicity against cancer cells.

**Figure 2 ijms-25-06848-f002:**
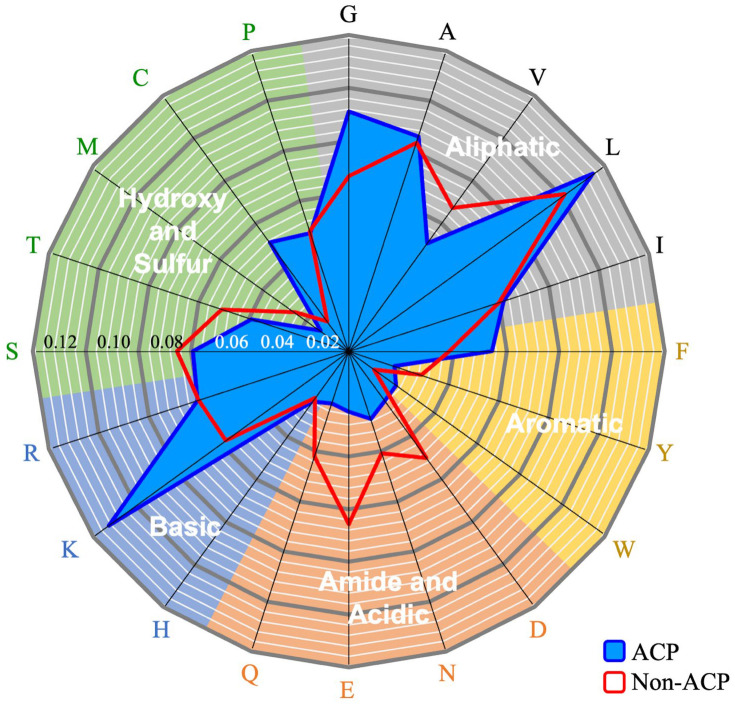
Comparison of amino acid compositions between ACPs and non-ACPs. The uppercase letters around the image represent various amino acids. The blue lines represent the amino acid compositions of ACPs, while the red lines represent those of non-ACPs.

**Figure 3 ijms-25-06848-f003:**
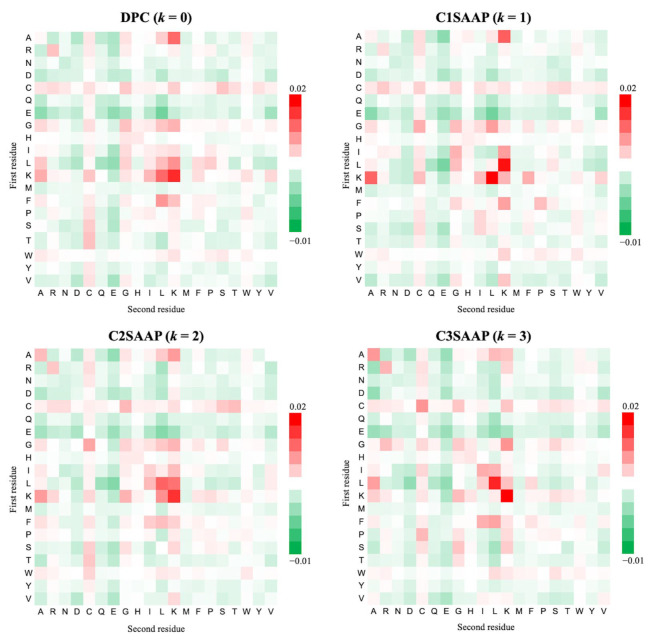
Comparison of frequencies of occurrence of 20 × 20 amino acid pairs separated by k residues between ACPs and non-ACPs.

**Figure 4 ijms-25-06848-f004:**
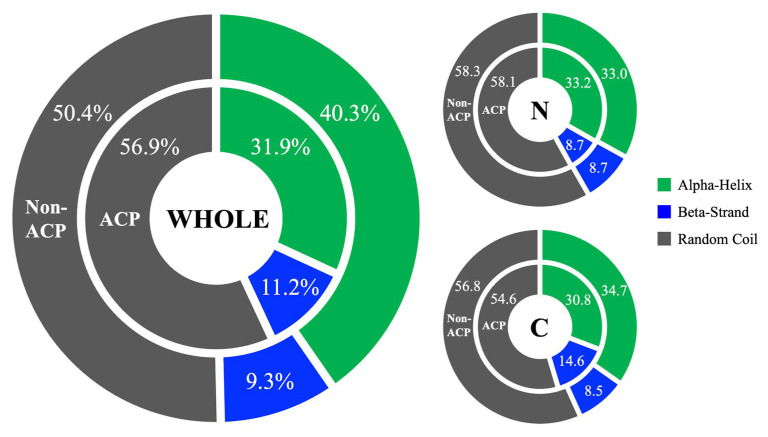
Comparison of compositions of secondary structure elements between ACPs and non-ACPs.

**Figure 5 ijms-25-06848-f005:**
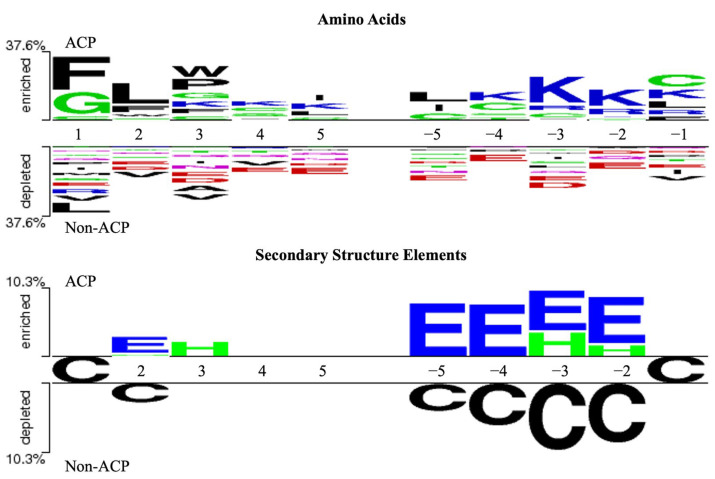
A comparison of the composition of amino acids and secondary structure elements at the N- and C-terminal regions between ACPs and non-ACPs. The uppercase letters in the upper part of figure represent the amino acids in the peptide sequences, with blue indicating positively charged amino acids and red indicating negatively charged amino acids. In the lower part, the uppercase letters represent the secondary structure of the peptide sequences, where H stands for alpha-helix, E stands for beta-sheet, and C stands for random coil.

**Figure 6 ijms-25-06848-f006:**
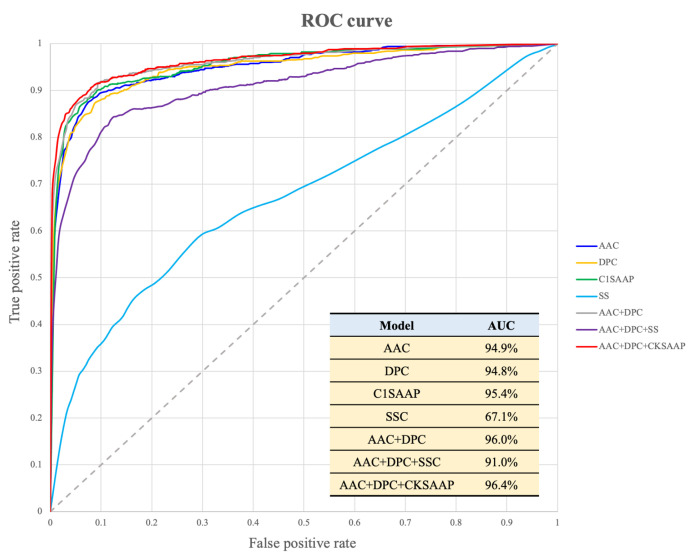
The ROC curves of models trained by sequence and structure-based features based on the results of five-fold cross-validation experiments.

**Figure 7 ijms-25-06848-f007:**
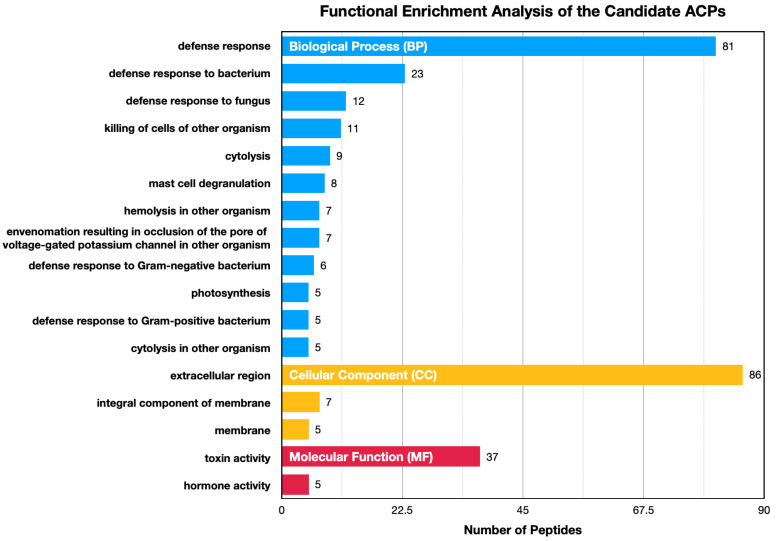
Functional enrichment analysis for candidate ACPs highlighting significant GO terms.

**Figure 8 ijms-25-06848-f008:**
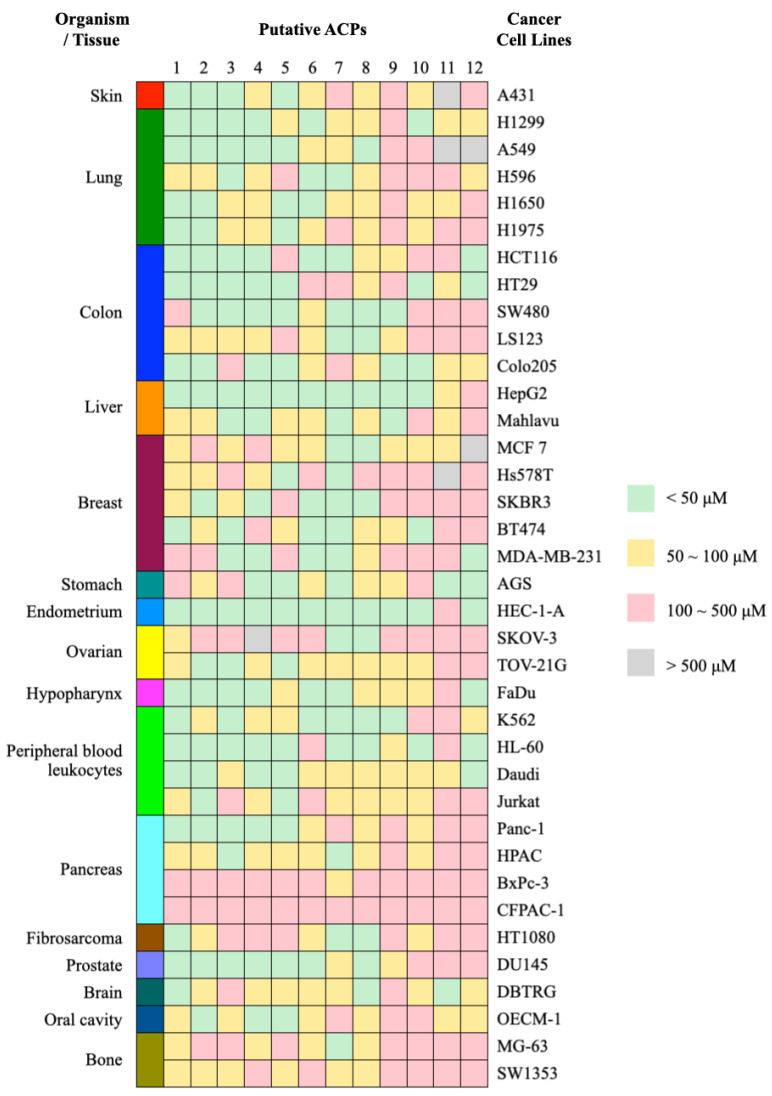
A validation of the anticancer activity of the putative ACPs against various cancer cell lines using the MTT colorimetric assay.

**Table 1 ijms-25-06848-t001:** Data statistics for the training and testing datasets.

Dataset	Number of ACPs	Number of Non-ACPs
Raw data	1462	2875
Length 10–50 aa	1344	2361
Training dataset	804	1494
Testing dataset	41,489 natural peptides	

**Table 2 ijms-25-06848-t002:** Results from five-fold cross-validation experiments for models trained with single feature sets.

Model	Sensitivity (%)	Specificity (%)	Accuracy (%)	MCC *
AAC	89.00 ± 0.0038	89.48 ± 0.0030	89.31 ± 0.0022	0.77 ± 0.0046
DPC	88.71 ± 0.0030	88.96 ± 0.0025	88.87 ± 0.0008	0.76 ± 0.0014
C1SAAP	90.00 ± 0.0017	90.09 ± 0.0035	90.06 ± 0.0022	0.79 ± 0.0042
C2SAAP	89.70 ± 0.0046	89.53 ± 0.0044	89.59 ± 0.0033	0.78 ± 0.0068
C3SAAP	89.10 ± 0.0014	90.58 ± 0.0025	90.06 ± 0.0016	0.79 ± 0.0033
SSEC	62.29 ± 0.0140	64.08 ± 0.0167	63.46 ± 0.0070	0.25 ± 0.0083
N-AAC	73.26 ± 0.0052	74.67 ± 0.0055	74.18 ± 0.0036	0.46 ± 0.0065
N-SSEC	50.92 ± 0.1771	50.67 ± 0.1782	50.76 ± 0.0539	0.02 ± 0.0027
C-AAC	69.18 ± 0.0077	70.08 ± 0.0048	69.77 ± 0.0032	0.38 ± 0.0069
C-SSEC	55.50 ± 0.0997	54.42 ± 0.1230	54.80 ± 0.0468	0.10 ± 0.0344

* MCC: Matthews correlation coefficient. The values represent the mean and standard deviation of all measurements.

**Table 3 ijms-25-06848-t003:** Results from five-fold cross-validation experiments for models trained with hybrid feature sets.

Model	Sensitivity (%)	Specificity (%)	Accuracy (%)	MCC *
AAC + DPC	91.02 ± 0.0025	90.12 ± 0.0047	90.44 ± 0.0037	0.80 ± 0.0074
AAC + DPC + SSEC	84.30 ± 0.0069	85.07 ± 0.0028	84.80 ± 0.0036	0.68 ± 0.0080
AAC + DPC + CKSAAP	91.17 ± 0.0037	90.83 ± 0.0032	90.95 ± 0.0021	0.81 ± 0.0043
AAC + DPC + CKSAAP + SSEC	86.52 ± 0.0047	87.07 ± 0.0033	86.88 ± 0.0015	0.72 ± 0.0030

* MCC: Matthews correlation coefficient. The values represent the mean and standard deviation of all measurements.

**Table 4 ijms-25-06848-t004:** Top 20 potential natural ACP candidates ranked by probability.

Entry Name	Sequence	ACPred	ACPred-FL	AntiCP	AntiCP2	iACP	mACPpred	Our Model
KAB4_OLDAF	GLPVCGETCVGGTCNTPGCTCSWPVCTRD	98.60%	98.11%	72.58%	96.00%	99.73%	98.17%	99.60%
CYO22_VIOOD	GLPICGETCVGGTCNTPGCTCSWPVCTRN	99.50%	95.12%	71.77%	95.00%	99.90%	98.42%	99.70%
THN2_VISAL	KSCCPNTTGRNIYNTCRFGGGSREVCASLSGCKIISASTCPSYPDK	99.50%	99.22%	70.56%	94.00%	99.52%	96.51%	96.60%
CYH3_VIOHE	GLPVCGETCFGGTCNTPGCICDPWPVCTRN	98.70%	92.89%	71.77%	95.00%	99.80%	98.80%	99.50%
MYX_CRODR	YKQCHKKGGHCFPKEKICIPPSSDFGKMDCRWRWKCCKKGSG	99.60%	99.22%	70.56%	91.00%	99.83%	94.71%	95.10%
KAB10_OLDAF	GLPTCGETCFGGTCNTPGCSCSSWPICTRD	99.40%	98.11%	70.56%	90.00%	99.93%	98.48%	99.60%
PROTO_POLPI	ILGTILGLLKSL	97.80%	99.22%	95.16%	54.00%	88.85%	98.35%	99.50%
CYO23_VIOOD	GLPTCGETCFGGTCNTPGCTCDSSWPICTHN	99.70%	98.11%	70.56%	87.00%	99.93%	98.47%	99.60%
CYPLE_PSYLE	SVTPIVCGETCFGGTCNTPGCSCSWPICTK	99.90%	99.22%	68.95%	87.00%	99.97%	96.86%	99.70%
CYPLD_PSYBR	GLPVCGESCFGGTCNTPGCSCTWPVCTRD	98.10%	95.12%	72.18%	87.00%	98.28%	98.01%	99.50%
ATOX_PHYTB	LTWKIPTRFCGVT	91.90%	99.22%	83.47%	50.00%	96.44%	96.64%	91.10%
CR12_RANCA	GLLGVLGSVAKHVLPHVVPVIAEHL	99.30%	98.11%	70.16%	84.00%	99.79%	98.62%	86.10%
KAB14_OLDAF	GLPVCGESCFGGTCNTPGCACDPWPVCTRD	88.70%	83.42%	71.77%	86.00%	99.76%	97.94%	99.00%
CYPLC_PSYLE	GDLPVCGETCFGGTCNTPGCVCAWPVCTR	95.70%	98.11%	68.15%	83.00%	99.25%	98.21%	99.40%
CYPLB_PSYLE	GDLPICGETCFGGTCNTPGCVCAWPVCNR	95.10%	98.11%	67.74%	83.00%	99.43%	97.87%	99.50%
GRAB_GRASX	IGGIISFFKRLF	100.00%	99.22%	85.08%	69.00%	82.43%	96.20%	98.90%
CIRF_CHAPA	AIPCGESCVWIPCISAAIGCSCKNKVCYR	99.60%	82.67%	75.81%	89.00%	99.80%	98.54%	99.50%
CYVNA_VIOIN	GIPVCGETCTLGTCYTAGCSCSWPVCTRN	99.60%	98.11%	71.37%	82.00%	99.80%	98.12%	99.50%
PNG1_PANCL	LNWGAILKHIIK	99.90%	99.22%	81.85%	58.00%	99.20%	98.22%	99.00%
PSMA3_STAAN	MEFVAKLFKFFKDLLGKFLGNN	98.60%	97.98%	75.81%	82.00%	96.15%	87.97%	85.70%

**Table 5 ijms-25-06848-t005:** List of putative ACPs selected for validation experiments.

Peptide	UniProt ID	Length	Sequence
ACP1	KAB4_OLDAF	29	GLPVCGETCVGGTCNTPGCTCSWPVCTRD
ACP2	CIRF_CHAPA	29	AIPCGESCVWIPCISAAIGCSCKNKVCYR
ACP3	PSMA3_STAAN	22	MEFVAKLFKFFKDLLGKFLGNN
ACP4	CYMEK_MELDN	31	GSIPCGESCVWIPCISSVVGCACKNKVCYKN
ACP5	CYVNA_VIOIN	29	GIPVCGETCTLGTCYTAGCSCSWPVCTRN
ACP6	CIRB_CHAPA	31	GVIPCGESCVFIPCISTLLGCSCKNKVCYRN
ACP7	THN2_VISAL	46	KSCCPNTTGRNIYNTCRFGGGSREVCASLSGCKIISASTCPSYPDK
ACP8	MYX_CRODR	42	YKQCHKKGGHCFPKEKICIPPSSDFGKMDCRWRWKCCKKGSG
ACP9	CR12_RANCA	25	GLLGVLGSVAKHVLPHVVPVIAEHL
ACP10	CYPLE_PSYLE	30	SVTPIVCGETCFGGTCNTPGCSCSWPICTK
ACP11	UT114_PEA	15	EQQQQQQPQNRRFRE
ACP12	TL11_SPIOL	22	FKGGGPYGQGVTRGQDLSGKDF

## Data Availability

The original data presented in the study are openly available at http://mer.hc.mmh.org.tw/datahub/dataset.php (accessed on 1 June 2024).

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
