# Peer review of "Integrating In Silico and In Vitro Approaches to Identify Natural Peptides with Selective Cytotoxicity against Cancer Cells"

_ijms, 2024, doi:10.3390/ijms25136848_

Round 1

Reviewer 1 Report

Comments and Suggestions for Authors

The present manuscript is dedicated to the identification of natural peptides with anti-cancer properties, called anti-cancer peptides (ACP), using an in silico approach. The pipeline used in this study was well described and the peptide prediction was convincing. However, some points should be added to improve this manuscript:

-          In an anti-cancer assay using some cancer cell lines from different cancers, the composition of the buffer dilution of the peptides was not mentioned. This composition should be reported.

-          The presence of % of fetal serum in the culture medium of cancer cell lines should be indicated.

-          For 1 or 2 peptides with the best effect on cancer cell lines, the anti-cancer dose-response curve should be shown. In addition, a clonogenic assay should be performed for these 1 or 2 peptides.

-          The anticancer effect of each ACP seems to depend on the cancer cell lines and/or the type of cancer. This should be discussed.

Reviewer 2 Report

Comments and Suggestions for Authors

Cancer is a major global health issue, causing millions of deaths annually. Traditional treatments like surgery and chemotherapy are often accompanied by severe side effects. Recently, anticancer peptides (ACPs) have emerged as promising alternatives due to their selective cytotoxicity towards cancer cells. These peptides, composed of 10-50 amino acids, disrupt cancer cell membranes through electrostatic interactions, exploiting the unique electrical properties of cancer cell surfaces. Computational methods have been developed to predict ACPs, utilizing features like amino acid composition and structural elements. Despite numerous predictive models, experimental validation remains limited. This study aims to combine computational predictions with experimental validation to identify natural peptides with anticancer properties, highlighting the potential of ACPs as effective and safer cancer therapies.

Author should address these questions:

  • How do the findings of this study impact the future development of cancer therapies?
  • What are the next steps for research in the field of ACPs, and how might these peptides be integrated into clinical practice?
Comments on the Quality of English Language

The manuscript is well-written with a high standard of English. The language is clear and concise, making the complex subject matter accessible to a broad audience. There are minor instances where sentence structure could be improved for better readability. Additionally, a few grammatical errors and typographical mistakes were noted, which can be easily corrected with careful proofreading. Overall, the quality of the English language is commendable, and with minor revisions, the manuscript will meet the high standards required for publication.

Round 2

Reviewer 1 Report

Comments and Suggestions for Authors

The authors have made changes to their manuscript in response to my earlier comments.